# Pretraining a Neural Operator in Lower Dimensions

**AmirPouya Hemmasian**                                    *hemmas@cmu.edu*
*Department of Mechanical Engineering*
*Carnegie Mellon University*

**Amir Barati Farimani**                                    *barati@cmu.edu*
*Department of Mechanical Engineering*
*Carnegie Mellon University*

**Reviewed on OpenReview:** *https: // openreview. net/ forum? id=ZewaRoZehI*

## Abstract

There has recently been increasing attention towards developing foundational neural Partial Differential Equation (PDE) solvers and neural operators through large-scale pretraining. However, unlike vision and language models that make use of abundant and inexpensive (unlabeled) data for pretraining, these neural solvers usually rely on simulated PDE data, which can be costly to obtain, especially for high-dimensional PDEs. In this work, we aim to **Pre**train neural PDE solvers on **Low**er **D**imensional PDEs (PreLowD) where data collection is the least expensive. We evaluated the effectiveness of this pretraining strategy in similar PDEs in higher dimensions. We use the Factorized Fourier Neural Operator (FFNO) due to having the necessary flexibility to be applied to PDE data of arbitrary spatial dimensions and reuse trained parameters in lower dimensions. In addition, our work sheds light on the effect of the fine-tuning configuration to make the most of this pretraining strategy. Code is available at https://github.com/BaratiLab/PreLowD.

## 1 Introduction

Many of the recent breakthroughs in artificial intelligence and deep learning are the fruits of large-scale pretrained models in all kinds of applications, ranging from computer vision (Chen et al., 2023), language processing (Devlin et al., 2018), prediction of molecular properties (Wang et al., 2022; Yu et al., 2021), and so on. Pretraining makes use of abundant data to learn useful and generalizable features and patterns that can be utilized in a downstream task. The effectiveness of using pretraining models is generally more significant when the downstream problem is complicated and/or the training data is scarce and expensive to collect (Chakraborty et al., 2022). In such cases, a simple model is prone to underfitting and a complex one to overfitting. Using a pretrained model can spare us the learning of basic and fundamental features from scratch and reduce the mentioned risks in the downstream task.

There are several core strategies for pretraining neural networks in different applications depending on the task and data at hand. For example, a common strategy in many computer vision tasks is to use a pretrained model on the ImageNet dataset with a simple task such as classification (He et al., 2019). Due to the large amount of data, this strategy has been beneficial in most computer vision tasks (Ridnik et al., 2021). This case of pretraining is done in a supervised learning framework, which requires a large amount of data that was expensive to collect but now is available. However, this is not the situation in many cases and that is where self-supervised learning comes into play.

Self-supervised learning (Liu et al., 2021) is similar to supervised learning with the difference that labels are automatically or trivially obtained from the data by itself without human input and are therefore inexpensive and fast to obtain. One of the most common and effective self-supervised pretraining strategies is masked autoencoding and prediction (Devlin et al., 2018; He et al., 2022; Wei et al., 2022; Song et al., 2020), in which the model has to predict missing parts or certain features or targets from a masked, noisy, or modified

version of the input. This strategy allows the model to extract and learn general patterns and features based on the context in which different parts of the data frequently appear. This framework is the key to the success behind many famous language models like BERT (Devlin et al., 2018) and ChatGPT (Wu et al., 2023). Image and video applications have also been shown to benefit from such strategies (He et al., 2022; Tong et al., 2022).

In addition to masked autoencoding and prediction tasks, other strategies have been introduced and proven effective. In general, any modification that maintains the semantics and central features of the data can be used as labels for a model to learn and thus extract generalizable features (Hu et al., 2019). These secondary tasks for self-supervised learning are also known as proxy tasks. Some examples of proxy tasks include sorting a shuffled input (scrambled words or image patches) (Kim et al., 2019; Panda et al., 2021), regressing a similarity or relevance score between pairs of data samples generated by augmenting or modifying identical or different raw samples (known as contrastive learning) (Hénaff et al., 2021; Rethmeier & Augenstein, 2023; Jaiswal et al., 2020), or similarly aligning the representation of encoding the same object from different data modalities (auditory, visual, and textual) (Lin et al., 2020; Wang et al., 2023). Due to the great novelty and success of all these strategies, neural PDE solvers and neural operators have also started to adapt the framework of pretrained models.

## 2 Related works

With the increasing availability of large PDE datasets and the success of pretraining strategies in classical applications, many researchers have been adapting such techniques for neural PDE solvers and neural operators. A simple question that might arise is whether it is effective to use a pretrained model trained on one PDE to learn the same PDE with different coefficients or different PDEs. To that point, Subramanian et al. (2024) investigated the transfer of the Fourier neural operator (FNO) to different coefficients, different PDEs, and how the result may vary with model scale and dataset size. They also tried pretraining the model on a mixed dataset consisting of several PDEs, which is the main focus of many following works. Some examples of such works include Multiple Physics Pretraining (MPP) by McCabe et al. (2023), Unified PDE Solvers (UPS) by Shen et al. (2024), Universal Physics Transformer (UPT) by Alkin et al. (2024), PDEFormer by Ye et al. (2024), and Denoising Pretrained Operator Transformer (DPOT) by Hao et al. (2024). These frameworks focus on designing a network architecture that is versatile enough to learn all PDEs in the training dataset, as well as other necessary techniques such as unifying the representation space and balancing the data feed between PDEs during training (Hao et al., 2024). The common characteristic of all these works is that the pretraining task and the downstream task are the same, that is, predicting the next time-step of the system (time-dependent PDE) or output approximation (boundary value problem).

Although not as popular as in classical applications, pretraining via proxy tasks has also been explored for neural PDE solvers, including elaborations on how to transfer and tune the pretrained models. Yang et al. (2023) defined a context-based prompt answering architecture, which they showed to be a few-shot PDE learner. Zhang et al. (2024) deciphered and integrated invariant features of physical systems in time and space in a contrastive learning framework and used them to fine-tune the first layer of a U-Net or FNO. Lorsung & Farimani (2024) used PDE coefficients to develop a more flexible pretraining approach based on contrastive learning. Tripura & Chakraborty (2023) selectively fine-tuned certain components of their designed architecture while preserving the pretrained weights of the wavelet-based components. Zhou & Farimani (2024) examined the capability of masked autoencoders as PDE learners, and later investigated a wide range of pretraining strategies for several neural operators and PDEs (Zhou et al., 2024). Mialon et al. (2023) proposed a contrastive learning strategy based on Lie symmetries in PDEs. Chen et al. (2024) applied classical pretraining strategies such as reconstructing blurred or masked inputs for neural PDE solvers. They used *unlabeled data* during pretraining, by which they meant the initial conditions in their datasets. Therefore, the pretraining stage actually relies on cheap data rather than expensive PDE datasets.

Up to this point, virtually all the cited works relied on a large pretraining dataset with a collection cost similar to that of the downstream task. Some focused on what one might rather call a multitask learning framework, while others looked into different ways to use the same kind of data with strategies to design proxy tasks or use and tune pretrained models. In contrast, we propose a novel pretraining approach to

increase the accuracy and pretraining efficiency of a neural operator for high-dimensional PDEs, that is, to pretrain the model in lower dimensions (1D). Since the cost of data collection and model training in 1D can be significantly lower than in 2D or 3D, there is a strong incentive to make use of this pretraining strategy. For example, to have a certain resolution $N$ per axis, a 1D and 2D system would be discretized to $N$ and $N^2$ points respectively. In traditional numerical solvers, each point represents an explicit update equation or an implicit equation to be solved. If the implicit solver relies on matrix inversion, it can increase the cost further to a power of 3, leading to a cost of $O(N^3)$ in 1D and $O(N^6)$ in 2D. In deep learning models, point-wise transformations are applied to each point, and field transformations (like those in the Fourier domain) have a cost that increases with the number of dimensions and the size of each dimension.

The pretraining strategy proposed in this work is not applicable to any arbitrary architecture or PDE, but when possible is much less costly than other pretraining approaches. In this work, we experiment with two fundamental PDEs that have a valid 1D and 2D version, and a neural operator that passes the requirement to be utilized in this framework. Since the fundamental blocks of PDEs are 1D derivatives and operators, this work is potentially a step towards building foundational models for solving PDEs as well. In our experiments, we observe that the average relative error of a prelowded factorized Fourier Neural Operator (FFNO) on the 2D diffusion equation can be 50% smaller than the same model trained from randomly initialized parameters over 5 rollout steps, which may not be achieved by simply increasing the amount of training data for the main training task. The gain of prelowding the model seems to amplify over more prediction time-steps and more rapidly changing systems (higher diffusion coefficient). Although the same results are not observed for the advection equation, this sheds light on the potential success of this approach in certain situations.

## 3 Background

### 3.1 Neural operators

Unlike the typical task of neural networks which is to approximate mappings between Euclidean spaces, neural operators approximate mappings between function spaces. An operator $\mathcal{G} : \mathcal{A} \to \mathcal{U}$ maps an input function $a \in \mathcal{A}$ to an output function $u \in \mathcal{U}$. In the application of solving PDEs, $a$ can include PDE coefficients and initial and boundary conditions, and $u$ is the solution of the PDE. In this work, we are considering time-dependent PDEs, where the input is the state of a physical system, and the output is the state at a later time. The operator then learns the mapping $u_t \to u_{t+\Delta t}$.

Although variables are defined and processed as functions, a discretized representation is eventually needed for numerical calculations. A higher resolution and a finer grid typically lead to more accurate results, but with the downside of higher computational cost and memory consumption, both for traditional numerical solvers and neural solvers. Typically, the cost increases exponentially with respect to the number of spatial axes. However, this is not the case for certain neural solvers that factorize the calculations across spatial axes. Some examples of such models are the Factorized Fourier Neural Operator (FFNO) (Tran et al., 2021), Axial Vision Transformer (AViT) (Ho et al., 2019; Bertasius et al., 2021) used in MPP (McCabe et al., 2023), and FactFormer (Li et al., 2024b;c). The central parameters and calculations of such models are defined per spatial axis, which makes the parameter count and computational cost the sum (rather than the product) of the single-axis amount, leading to a linear cost with respect to the number of spatial axes instead of an exponential cost. Moreover, it is possible to reuse the parameters of a pretrained 1D model when learning to predict a relevant system in higher dimensions. Our quest in this work is to investigate the potential of a model Pretrained in Lower Dimensions (PreLowDed) to be fine-tuned in higher dimensions. We choose to explore this strategy with the Factorized Fourier Neural Operator (FFNO) (Tran et al., 2021) because of its efficiency due to the fast Fourier transform (FFT) and factorization. Following works can look into composition and factorization of other attention-based architectures like Galerkin Transformer (Cao, 2021) and OFormer (Li et al., 2022), as well as using this strategy in a learned encoding space (Hemmasian & Barati Farimani, 2023; Hemmasian & Farimani, 2024; Li et al., 2024a).

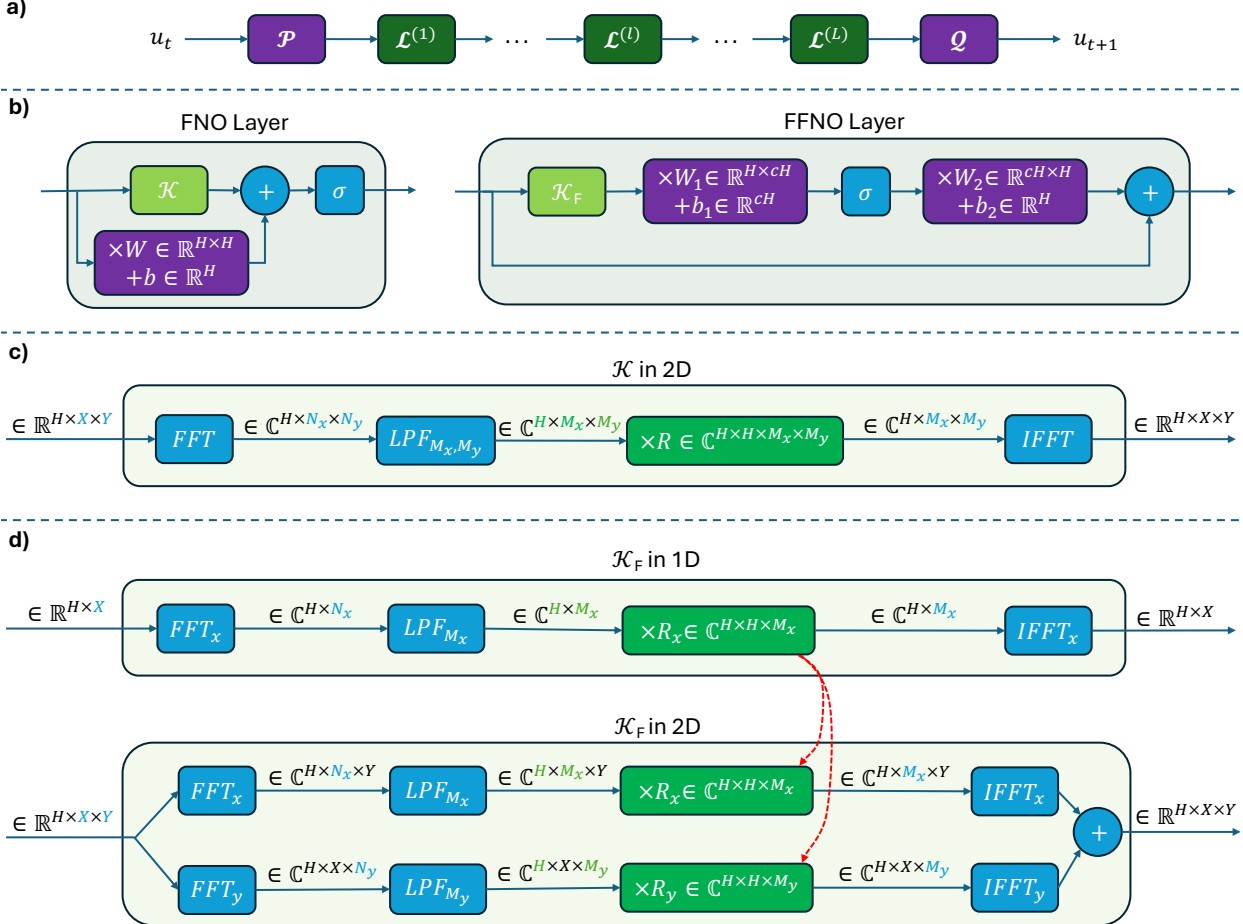

Figure 1: a) General schematic of a neural operator. b) The nonlinear operator layer in FNO and FFNO. c) The kernel integral operator in FNO. $LPF$ stands for a low pass filter that keeps the first few Fourier modes in each axis and discards the higher frequency modes. d) 1D and 2D factorized kernel integral operator in FFNO. The red arrows show the possible transfer of pretrained parameters from a 1D model to a 2D model if $M_x = M_y$.

### 3.2 Factorized Fourier Neural Operator

The original Fourier Neural Operator (FNO) was introduced by Li et al. (2020a) in a line of work based on an architecture analogous to typical neural networks. By replacing linear layers in a neural network with linear operators, Li et al. (2020b) proposed an architecture template shown in figure 1a to construct a neural operator through iterative linear and nonlinear transformations of the input:

$$u = \mathcal{G}(a) = \left( \mathcal{Q} \circ \mathcal{L}^{(L)} \circ \cdots \circ \mathcal{L}^{(1)} \circ \mathcal{P} \right)(a), \tag{1}$$

where $\circ$ is function composition, $L$ is the number of layers, $\mathcal{P}$ is the input projector from the physical space to the latent space, $\mathcal{L}^{(l)}$ is the $l$th non-linear operator layer, and $\mathcal{Q}$ is the output projector from the latent space back to the physical space. A nonlinear operator layer of FNO executes the following transformations on the latent representation:

$$\mathcal{K}^{(l)}\left( z^{(l-1)} \right) = \mathrm{IFFT}\left( R^{(l)} \cdot \mathrm{FFT}(z^{(l-1)}) \right) \tag{2}$$

$$z^{(l)} = \mathcal{L}^{(l)}\left( z^{(l-1)} \right) = \sigma\left( W^{(l)} z^{(l-1)} + b^{(l)} + \mathcal{K}^{(l)}(z^{(l-1)}) \right) \tag{3}$$

where $\mathcal{K}^{(l)}$ is a kernel integral operator that executes a convolution integral via matrix multiplication in the Fourier space, $R_d^{(l)}$ is the Fourier weight matrix, $\sigma : \mathbb{R} \to \mathbb{R}$ is a point-wise nonlinear activation function, and $W^{(l)} z^{(l-1)} + b^{(l)}$ is an affine point-wise map in physical space. You can see a simple illustration of the FNO layer on the left in figure 1b, and a 2D kernel integral operator block in figure 1c. Assuming the same number of Fourier modes in each axis ($M_x = M_y = M$), each layer has a distinct or shared complex weight matrix $R^{(l)}$ containing $H^2 M^D$ complex parameters, where $H$ represents the hidden or latent dimension (also known as width) and $D$ is the number of spatial axes.

Later, Tran et al. (2021) proposed a factorized version of FNO (FFNO) with a factorized kernel integral operator $\mathcal{K}_F$ and a modified layer $\mathcal{L}^{(l)}$:

$$\mathcal{K}_F^{(l)}\left( z^{(l-1)} \right) = \sum_d \left[ \mathrm{IFFT}_d\left( R_d^{(l)} \cdot \mathrm{FFT}_d\left( z^{(l-1)} \right) \right) \right] \tag{4}$$

$$z^{(l)} = \mathcal{L}^{(l)}\left( z^{(l-1)} \right) = z^{(l-1)} + W_2^{(l)} \sigma\left( W_1^{(l)} \mathcal{K}_F^{(l)}\left( z^{(l-1)} \right) + b_1^{(l)} \right) + b_2^{(l)} \tag{5}$$

Here, $\mathcal{K}_F$ is a factorized kernel integral operator, $R_d^{(l)}$ is the Fourier weight matrix of axis $d$, and $\sigma, W, b$ represent the activation, weights, and biases of the feedforward layers respectively. A simple visualization of the FFNO layer and its factorized kernel integral operator are provided in figure 1b and 1d respectively. FFNO fully preserves the residual connection throughout the layers to keep the original input and its information as much as possible, and applies the nonlinearity to the output of the kernel integral instead. It also uses a two-layer feedforward layer instead of the single-layer one in FNO, which is inspired by the transformer architecture (Vaswani et al., 2017). The central feature of FFNO is the factorized kernel integral operator whose cost and parameters count in each dimension is $O(H^2 M)$, summing up to $O(H^2 M D)$.

Not only does the factorized architecture save a lot of computational cost and memory usage when storing, training, and testing the model, but it also allows weights to be of the same shape across different axes if the number of modes is the same. This opens up the possibility of transferring weights across different problems with an arbitrary number of axes. In this work, we evaluate the effectiveness of this strategy on the advection and diffusion equations, two simple but fundamental PDEs.

## 4 Methodology

Before the formal definition of how to prelowd a neural PDE solver or operator, a couple of requirements must be met. First, the architecture of the neural operator has to be compartmentalized per spatial axis, which is usually indicated by it being called an axial or factorized neural solver or operator. This enables the generalization and recycling of pretrained parameters of an axial module to other axes in the high-dimensional PDE of the downstream task. The next requirement is the relevance of the pretraining PDE,

which is defined in a space with fewer dimensions. Some terms in PDEs such as the gradient or the Laplacian, or their nonlinear combination with other terms, can be defined and appear in spaces with different numbers of dimensions. The existence of such similar terms in a PDE is a reasonable justification to use it to construct the pretraining dataset. The step-by-step pipeline of prelowding a factorized or axial neural PDE solver can be found in Algorithm 1.

---

**Algorithm 1** PreLowDing a neural PDE solver

---

**Input:** a neural PDE solver and a target (downstream) PDE dataset to learn
Step 1: Verify the neural solver has a factorized/axial architecture.
Step 2: Choose a relevant low-dimensional (1D) PDE for pretraining. A relevant PDE would consist of similar terms and similar physical phenomena (advection, diffusion, etc) in the lower dimensions. Construct a training dataset from the chosen PDE(s).
Step 3: Train the neural operator on the pretraining dataset. This step is where the pretraining is executed.
Step 4: After training, replicate the axial module(s) of the model to the number of dimensions in the downstream (target) PDE.
Step 4.5: Apply any appropriate modification before the neural solver is trained (fine-tuned) on the downstream dataset. An example of such modification is freezing certain components not to be changed in the fine-tuning stage.
**Output:** The prelowded model ready to be trained (fine-tuned) on the downstream PDE

---

Moving on, we will continue the framework focusing on FFNO as our choice of neural PDE solver. As explained previously and illustrated in figure 1d, a 1D FFNO and a 2D FFNO can share all their corresponding parameters if the number of preserved Fourier modes is the same in x and y. The projection layers and the feedforward layers are point-wise transformations in the physical space, and all Fourier weights are defined for a single axis. After transferring the pretrained parameters, we explore different configurations to freeze or fine-tune them for the downstream task. In classical applications like computer vision, the typical strategy is to freeze the weights of the initial layers (also known as the stem) and only tune the parameters of the final layer(s). This is justified due to the hierarchical nature of the architecture and the learned features. Moreover, tuning the model is cheaper since the backpropagation is not executed all the way back to the input, and the sheer number of trainable parameters is also decreased by freezing the stem of the model. In addition, the scarcity of training data in the downstream task introduces the risk of overfitting, which can be mitigated by this strategy due to the low number of trainable parameters. Keeping this in mind, we have to decide how to choose the best way to perform the fine-tuning considering the architecture of neural operators and PDE data.

The first difference that we pay attention to is the absence of a hierarchical architecture and feature extraction in models like FNO and FFNO. Unlike convolutional neural networks that have a shrinking architecture towards the output, these neural operators maintain the same resolution and stay in roughly the same latent space throughout their hidden layers. Therefore, the classic fine-tuning strategies of classic applications may not be the best choice here. However, these models consist of several types of components, each of which may learn certain properties and features of the system. For example, the parameters of FFNO belong to either projectors, kernel integral operator layers, or feedforward layers. We can choose to fine-tune only a subset of the model's components and see how the tuned model performs in the downstream task. This can also pave the path toward the interpretability of such models by breaking down the information learned by each component.

In addition to choosing the tunable subset of the model based on component type, we also consider fine-tuning based on the chronological order of the layers. This includes not only the classical approach of fine-tuning the last layer(s), but also fine-tuning the first layer(s) or a combination of both. If we want to try all the possible configurations, the number of experiments exponentially increases, since we may choose to tune or not tune each component of each layer. Therefore, we narrow our options down to eight configurations as shown in table 1. We will represent the model without pretraining as C0. The projector layers $\mathcal{P}, \mathcal{Q}$ are left to be tuned in all cases. The configurations C2 to C8 were selected so the tunable modules are either in proximity to the model's input or output , or are of the same type (Fourier or feedforward). While C1

represents a fully tunable model, C2 and C3 have a certain tunable layer type across all layers and the remaining configurations have one or both layer types from the first and/or last layer.

Table 1: The trainable components of the fine-tuning configurations.

| parameters | C1 | C2 | C3 | C4 | C5 | C6 | C7 | C8 |
|---|---|---|---|---|---|---|---|---|
| $R^{(0)}$ | ✓ | ✓ | × | × | ✓ | ✓ | ✓ | × |
| $W^{(0)}, b^{(0)}$ | ✓ | × | ✓ | × | × | ✓ | × | × |
| $R^{(l)}(0 < l < L)$ | ✓ | ✓ | × | × | × | × | × | × |
| $W^{(l)}, b^{(l)}(0 < l < L)$ | ✓ | × | ✓ | × | × | × | × | × |
| $R^{(L)}$ | ✓ | ✓ | × | × | × | × | × | ✓ |
| $W^{(L)}, b^{(L)}$ | ✓ | × | ✓ | ✓ | × | × | ✓ | ✓ |

We will evaluate the performance of the models by comparing the next-step prediction error and the average error over a 5-step autoregressive rollout. We also conduct experiments with different numbers of training samples in the downstream task. As we shall see, the advantage of pretraining depends on the availability of training data in the downstream task.

## 5 Experiments

### 5.1 Datasets and objectives

The general format and domains of the PDEs in this work are shown in equation 6, where $N$ is a linear or nonlinear function, $\mathbf{c}$ is the vector of the PDE coefficients, $u$ is the quantity of interest and $t, \mathbf{x}$ are the time and space variables, respectively. Each sample of a dataset consists of the evolution of the system in the specified time domain sampled every $\Delta t = 0.05$, resulting in 21 snapshots and 20 input-output pairs on which the model will be trained. The spatial resolutions for the 1D and 2D datasets are set to 1024 and $64^2$, respectively. Different samples of each dataset are governed by the same equation and coefficients, and differ only in the initial condition defined by equation 7 where $\mathbf{A}_i, n_i, \phi_i$ are random variables drawn from uniform distributions, similar to the multi-dimensional datasets in PDEBench (Takamoto et al., 2022). We assume periodic boundary conditions in all cases.

$$u_t = N(\mathbf{c}, u, u_x, u_{xx}, u_{xy}, ...) \tag{6}$$
$$u = u(\mathbf{x}, t) \in \mathbb{R}, \quad t \in [0, 1], \quad \mathbf{x} \in (0, 1)^D, \quad D \in \{1, 2\}$$

$$u_0(\mathbf{x}) = u(\mathbf{x}, t = 0) = \sum_{i=1}^{N} \mathbf{A}_i sin(2\pi n_i \mathbf{x} + \phi_i) \tag{7}$$

For each dimensionality and each value of $\mathbf{c}$, 10000 samples were generated, 2000 of which are held for validation. Since the characteristics of a typical scenario for pretraining a model are the high collection cost and scarcity of data, we set up the experiments to resemble such scenarios. For each case (a particular PDE with a particular value for the coefficient), we first pretrain the model on the relevant 1D pretraining dataset. Then, we fine-tune the pretrained model with different amounts of available downstream training data. That is to find out the relationship between downstream dataset size and the gain of pretraining compared to training from scratch. To clarify, the experiment for each PDE and each value for the coefficient(s) of the PDE are separate and independent of each other. For example, if the downstream task is to learn 2D diffusion with a diffusion coefficient of $\nu = 0.002$, the pretraining dataset consists only of samples from 1D diffusion equation with the same diffusion coefficient of $\nu = 0.002$.

For the downstream task, we compare a model trained from random initialization and 8 pretrained models with fine-tuning configurations shown in table 1. The objective of both training and validation is the relative $L_2$ norm, also known as nRMSE (Takamoto et al., 2022), of the prediction error shown in equation 8 where $\hat{y}, y$ are the model output and target value for the quantity of interest, respectively. In our application, $y$ is the state of the system at the next time-step. We train the model on the next-step prediction loss and evaluate the model using the same metric on the validation dataset, as well as the average loss over a 5-step

autoregressive rollout similar to what was done in PDEBench (Takamoto et al., 2022). Finally, we run each experiment with 3 different random seeds and report the average result to take into account the randomness of the network initialization, as well as the randomly selected subset of the 2D training dataset. Now, we will look at the specifics of each equation, the result for each dataset, and discuss the effect of the fine-tuning strategy, the number of training samples, and the PDE coefficient based on the observed results. We will continue the discussion here using concise plots. Please refer to Appendix A for detailed numerical tables on all the curated results.

$$rL_2(\hat{y}, y) = \frac{||\hat{y} - y||_2}{||y||_2} \tag{8}$$

Each training stage consists of 5000 iterations of the AdamW optimizer with an initial learning rate of 0.001. The learning rate is multiplied by 0.2 when the loss reaches a plateau and does not improve for more than 100 iterations. Our choice of architecture for FFNO has 4 hidden layers with a latent dimension of 128 and 16 Fourier modes in each axis.On our GeForce RTX 2080 Ti Nvidia GPUs, with a batch size of 64 and 5000 optimization iterations for each stage, the pretraining takes about 3.35 minutes while the fine-tuning or training for the 2D task takes about 21 minutes. This is an example of how the training cost differs for a 2D PDE compared to 1D. This is despite the resolution in 2D being 64 while the 1D data has a resolution of 1024.

## 5.2 Diffusion equation

We generated six different diffusion datasets by solving equation 9 in 1D and 2D for each coefficient. An implicit Euler scheme with $\Delta t = 0.001$ was utilized.

$$u_t = \nu\nabla^2 u, \quad \nu \in \{0.001, 0.002, 0.004\} \tag{9}$$

The results for the diffusion datasets are shown in figure 2. The x-axis represents the number of downstream training samples, and the y-axis represents the average loss on the 2000 validation samples. We exclude the instances trained with only 1 or 2 training samples due to their unacceptable amount of error. It appears that the fine-tuning configurations C1, C2, and C8 perform as well or better than the randomly initialized model C0, and the remaining configurations fail to achieve the same error as C0. Focusing on C1 (all parameters trainable) and C0, the first trivial indicator of success is that C1 outperforms C0 and reduces the error up to 80% in the low-data regime. The second interesting observation is the different trend of the error with respect to the number of training parameters. The error of C0 starts from a very high value in the left side with very few samples, and improves significantly with the increase of the training dataset size. However, the performance of C1 even with very few samples on the left is already comparable to the performance on the right end of the plot. An interesting observation is that C1 trained on 8 samples outperforms C0 trained on 1024 samples for $\nu = 0.004$. There we can also see more clearly that the advantage of pretraining is amplified over autoregressive rollout. For exact values, refer to Appendix A.

The second trend that we discuss is how the advantage varies for different coefficient values of the PDE. In the diffusion equation, the larger coefficient means a more rapidly changing system. It can be seen in figure 2 that for a larger dataset size, the differences between C1 and C0 are relatively small for $\nu = 0.001$. However, the gap becomes more significant as we increase the diffusion coefficient. For the 5-step rollout for $v = 0.004$, C1 still achieves a 40-50% error reduction compared to C0. Statistically speaking, the distribution of the system state remains relatively more familiar over time if the model changes less rapidly. Therefore, more unfamiliar samples and a different state distribution are presented to the model during validation if the system has a rapid rate of change, further increasing the risk of overfitting and poor performance on the unseen validation data. However, the pretrained models have already seen a large amount of data, hence mitigating this risk to some extent.

Focusing on the fine-tuning configurations, C1, C2, and C8 are the models that perform best. The explanation is trivial for C1, since it is totally free to be tuned on the downstream data. However, it is interesting that C8 with only the last layer left trainable outperforms C2 (all Fourier layers trainable) despite having fewer trainable parameters. Each Fourier and feedforward layer have about 524K and 60K parameters, respectively. This means that C8 with less than 600K trainable parameters performs as well as or better than C2 with

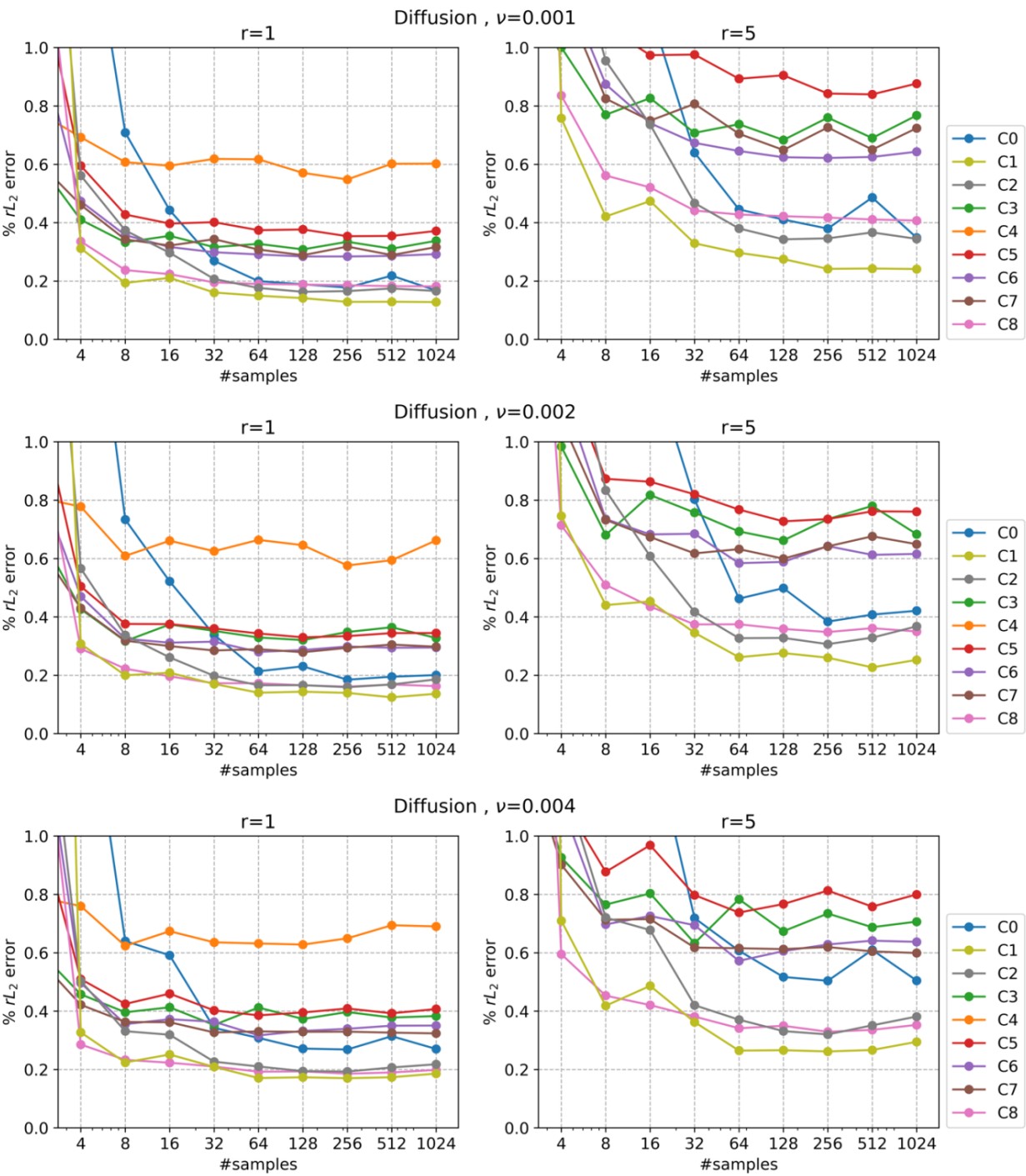

Figure 2: Average $rL_2$ loss in percentage for the diffusion equation. C0 is the randomly initialized model and the rest are PreLowDed models fine-tuned according to table 1. The left column shows the next-step error (rollout=1) and the right column shows the average error over the next five autoregressive steps (rollout=5).

more than 2 million trainable parameters. This can be indicative of a hierarchical order of the learning mechanism of the model for the diffusion equation. In the extreme low-data regime (less than 10), tuning configurations with less trainable parameters appear to have the most advantage according to the tables in Appendix A. This makes sense since having fewer trainable parameters reduces the risk of overfitting. However, they fall behind when more training samples are available.

### 5.3 Advection equation

Similarly to the diffusion dataset, six advection datasets were provided for our experiments. The 1D datasets are provided by PDEBench (Takamoto et al., 2022), and we generated the 2D datasets using the exact solution function of $u(x, y, t) = u_0(x - \beta t, y - \beta t)$.

$$u_t = -\beta \nabla u, \quad \beta \in \{0.1, 0.4, 1.0\} \tag{10}$$

For the advection equations, the strategy does not seem to be as successful as the diffusion equations. Based on the results provided in figure 3, the difference between the best pretrained models and C0 is significant only to the far left of the plot with very few samples. The two fine-tuning configurations that seem to be on par with C0 are C1 and C2, which are the ones with the most number of trainable parameters. Unlike the case for diffusion equations, C8 performs poorly, falling into the second last model for $\beta = 1.0$, which can indicate a fundamental difference in how the model extracts features and learns to solve the advection equation compared to the diffusion equation. With an increase of $\beta$, the rate of change increases, and the prediction task becomes more difficult. Unlike the diffusion equations where pretraining showed even more benefit over autoregressive rollout, there does not seem to be any significant difference for the advection equations.

## 6 Conclusion

This work proposes a novel strategy to reduce the cost of data collection, training, and error of a neural operator in multidimensional PDEs. For a neural operator with a factorized architecture like FFNO, we can pretrain a model in lower dimensions (PreLowD) and transfer the weights to a multidimensional model. We showed that this strategy is successful for the 2D diffusion equation, but not for the advection equation. When successful, the PreLowDed model outperformed the randomly initialized model by up to 80% in low-data regimes and a slow-changing system. When the system has a faster rate of change or the model is used autoregressively, the gain of the PreLowDed model is also very significant, achieving an error reduction of 50% over a 5-step rollout even with 1024 training samples.

We explored several fine-tuning strategies to find the best way to freeze or tune the transferred weights from a PreLowDed model. It seems that with sufficient training, more trainable parameters will result in a lower error. However, tuning a smaller subset of the parameters may achieve a lower error when very few training samples are available. For the advection equation, the only important factor seemed to be the sheer number of tunable parameters. However, tuning the final full layer for the diffusion datasets was among the best choices, indicating a possibly hierarchical feature extraction of the model. Our general recommendation is to fine-tune the final layer in case of extreme scarcity of training data in the downstream task, or to fine-tune the whole model with all parameters being trainable.

In future work, this pretraining strategy may be utilized with other neural operators and PDE solvers that have the necessary properties as well. However, it may not be straightforward to find or define a similar physical system with fewer number of dimensions for pretraining. Moreover, the selective fine-tuning strategy can be applied to any neural operator and can provide insight into the generalizability and interpretability of such models. For example, tuning different parts of the model while varying a certain coefficient or term for the PDE of the downstream task can help discover the correspondence of the neural operator's components and different modalities and terms of the PDE. Finally, an important potential venue for research is the study of different possible compositions of low-dimensional models and factorization of high-dimensional models. The ultimate goal of this direction is to improve the prediction accuracy and efficiency in more complex or 3D systems, and this work was just a humble starting step towards that end.

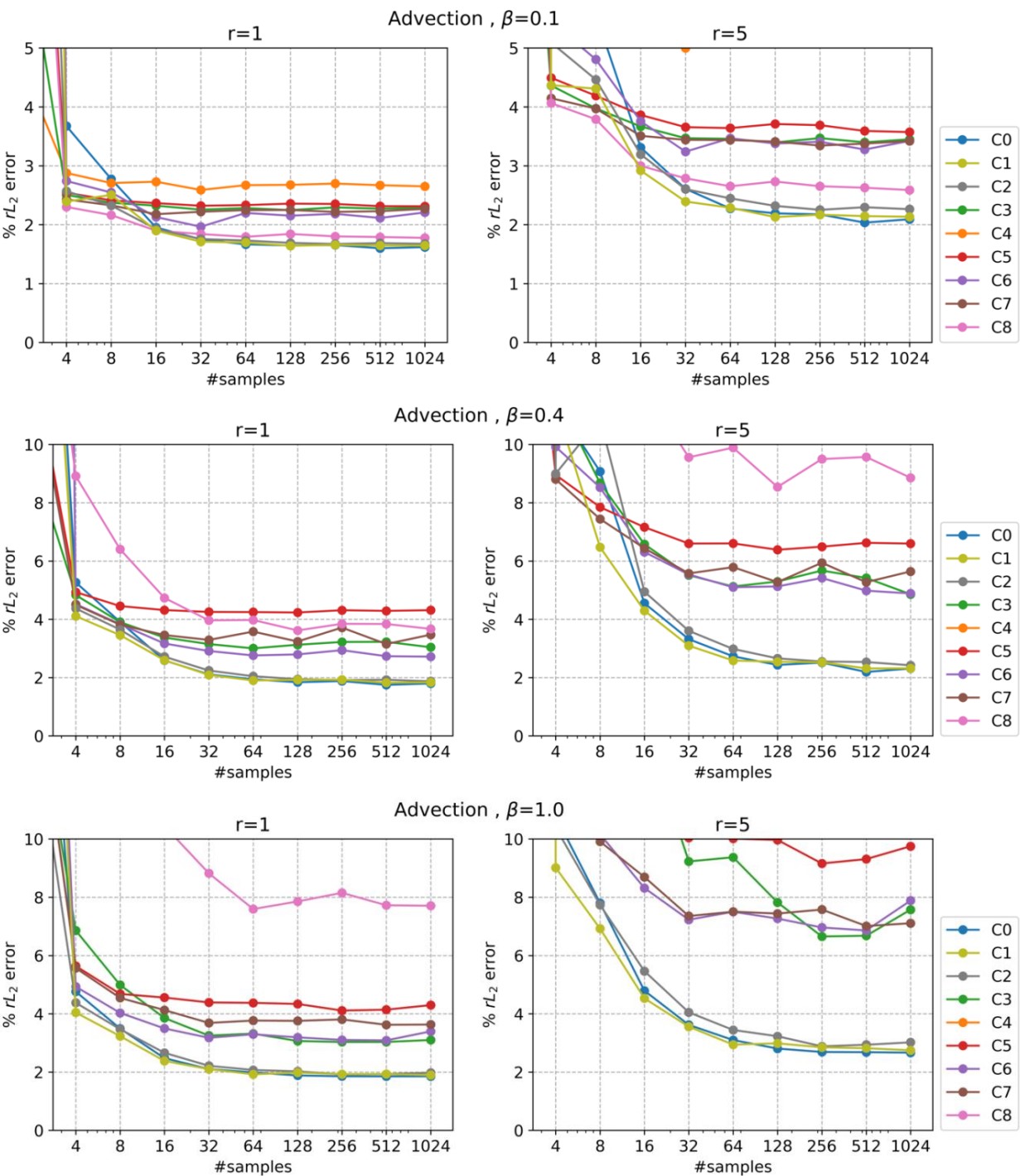

Figure 3: Average $rL_2$ loss in percentage for the advection equation. C0 is the randomly initialized model and the rest are PreLowDed models fine-tuned according to table 1. The left column shows the next-step error (rollout=1) and the right column shows the average error over the next five autoregressive steps (rollout=5).

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

## A  Appendix

All the results are presented in detail in this section. The reported metrics are the next-step prediction error and the average error over the next five autoregressive rollout steps of the model. C0 is the model trained from a random initialization and the rest are pretrained models with different fine-tuning configurations as specified in table 1.

Table 2: **Results for Diffusion** $\nu = 0.001$. Average $rL_2$ loss in percentage, and the relative change compared to C0 (no pretraining) in parenthesis. Negative changes indicate improvement.

| #samples | rollout | C0 | C1 | C2 | C3 | C4 | C5 | C6 | C7 | C8 |
|---|---|---|---|---|---|---|---|---|---|---|
| 1 | r=1 | 15.42 | 6.08(-60.6%) | 3.79(-75.4%) | 0.93(-93.9%) | 1.06(-93.1%) | 1.66(-89.2%) | 1.55(-89.9%) | 0.99(-93.6%) | 1.84(-88.1%) |
| | r=5 | 30.55 | 14.41(-52.8%) | 13.33(-56.4%) | 2.55(-91.6%) | 3.06(-90.0%) | 5.05(-83.5%) | 4.73(-84.5%) | 2.81(-90.8%) | 5.08(-83.4%) |
| 2 | r=1 | 9.82 | 3.26(-66.8%) | 2.37(-75.8%) | 0.62(-93.7%) | 0.78(-92.0%) | 1.32(-86.6%) | 1.05(-89.3%) | 0.62(-93.7%) | 1.67(-83.0%) |
| | r=5 | 17.43 | 7.42(-57.4%) | 8.23(-52.8%) | 1.68(-90.4%) | 2.23(-87.2%) | 4.09(-76.5%) | 3.13(-82.0%) | 1.70(-90.2%) | 4.24(-75.7%) |
| 4 | r=1 | 1.67 | 0.31(-81.3%) | 0.56(-66.4%) | 0.41(-75.5%) | 0.69(-58.5%) | 0.59(-64.4%) | 0.47(-71.7%) | 0.46(-72.4%) | 0.34(-79.9%) |
| | r=5 | 4.08 | 0.76(-81.5%) | 1.50(-63.3%) | 1.00(-75.5%) | 1.93(-52.8%) | 1.60(-60.9%) | 1.25(-69.5%) | 1.19(-70.9%) | 0.84(-79.5%) |
| 8 | r=1 | 0.71 | 0.19(-72.7%) | 0.37(-47.3%) | 0.33(-53.1%) | 0.61(-14.3%) | 0.43(-39.6%) | 0.36(-49.6%) | 0.34(-51.7%) | 0.24(-66.5%) |
| | r=5 | 1.91 | 0.42(-77.9%) | 0.95(-49.9%) | 0.77(-59.6%) | 1.67(-12.4%) | 1.07(-43.7%) | 0.87(-54.1%) | 0.82(-56.7%) | 0.56(-70.5%) |
| 16 | r=1 | 0.44 | 0.21(-52.4%) | 0.30(-33.0%) | 0.35(-19.9%) | 0.60(+34.3%) | 0.40(-10.4%) | 0.32(-28.5%) | 0.32(-27.4%) | 0.22(-49.5%) |
| | r=5 | 1.13 | 0.47(-57.9%) | 0.74(-34.7%) | 0.83(-26.6%) | 1.62(+44.0%) | 0.97(-13.6%) | 0.74(-34.2%) | 0.75(-33.5%) | 0.52(-53.8%) |
| 32 | r=1 | 0.27 | 0.16(-40.3%) | 0.21(-23.0%) | 0.32(+17.6%) | 0.62(+130.0%) | 0.40(+49.3%) | 0.30(+11.0%) | 0.34(+27.6%) | 0.20(-27.4%) |
| | r=5 | 0.64 | 0.33(-48.6%) | 0.47(-27.0%) | 0.71(+10.6%) | 1.70(+165.5%) | 0.98(+52.5%) | 0.67(+5.2%) | 0.81(+26.1%) | 0.44(-31.0%) |
| 64 | r=1 | 0.20 | 0.15(-25.0%) | 0.18(-11.8%) | 0.33(+63.9%) | 0.62(+209.1%) | 0.37(+87.4%) | 0.29(+45.8%) | 0.31(+54.2%) | 0.19(-4.8%) |
| | r=5 | 0.45 | 0.30(-33.4%) | 0.38(-14.7%) | 0.74(+65.5%) | 1.70(+280.9%) | 0.89(+100.5%) | 0.65(+44.9%) | 0.70(+58.2%) | 0.43(-3.9%) |
| 128 | r=1 | 0.19 | 0.14(-25.1%) | 0.16(-13.6%) | 0.31(+63.4%) | 0.57(+202.4%) | 0.38(+99.6%) | 0.28(+50.6%) | 0.29(+52.8%) | 0.19(-0.5%) |
| | r=5 | 0.41 | 0.28(-33.0%) | 0.34(-16.5%) | 0.68(+66.4%) | 1.55(+277.7%) | 0.90(+120.4%) | 0.62(+52.0%) | 0.65(+58.0%) | 0.42(+2.8%) |
| 256 | r=1 | 0.18 | 0.13(-27.6%) | 0.17(-6.8%) | 0.33(+88.6%) | 0.55(+208.9%) | 0.35(+99.3%) | 0.28(+60.1%) | 0.32(+79.3%) | 0.19(+4.6%) |
| | r=5 | 0.38 | 0.24(-36.3%) | 0.35(-8.8%) | 0.76(+100.2%) | 1.48(+289.4%) | 0.84(+121.9%) | 0.62(+63.8%) | 0.73(+91.3%) | 0.42(+9.9%) |
| 512 | r=1 | 0.22 | 0.13(-40.9%) | 0.17(-20.0%) | 0.31(+42.4%) | 0.60(+175.8%) | 0.35(+62.4%) | 0.29(+31.1%) | 0.29(+32.4%) | 0.18(-16.4%) |
| | r=5 | 0.49 | 0.24(-50.0%) | 0.37(-24.6%) | 0.69(+42.0%) | 1.65(+238.9%) | 0.84(+72.7%) | 0.63(+28.6%) | 0.65(+33.8%) | 0.41(-15.5%) |
| 1024 | r=1 | 0.17 | 0.13(-23.7%) | 0.17(-1.1%) | 0.34(+101.6%) | 0.60(+259.4%) | 0.37(+121.8%) | 0.29(+74.3%) | 0.32(+88.8%) | 0.18(+8.5%) |
| | r=5 | 0.35 | 0.24(-30.9%) | 0.34(-1.2%) | 0.77(+120.2%) | 1.65(+373.4%) | 0.88(+151.5%) | 0.64(+84.5%) | 0.72(+107.7%) | 0.41(+16.9%) |

Table 3: **Results for Diffusion** $\nu = 0.002$. Average $rL_2$ loss in percentage, and the relative change compared to C0 (no pretraining) in paranthesis. Negative changes indicate improvement.

| #samples | rollout | C0 | C1 | C2 | C3 | C4 | C5 | C6 | C7 | C8 |
|---|---|---|---|---|---|---|---|---|---|---|
| 1 | r=1 | 17.29 | 7.45(-56.9%) | 3.21(-81.4%) | 0.97(-94.4%) | 1.08(-93.8%) | 1.84(-89.4%) | 1.61(-90.7%) | 0.87(-95.0%) | 1.37(-92.0%) |
| | r=5 | 32.67 | 16.72(-48.8%) | 10.53(-67.8%) | 2.68(-91.8%) | 3.09(-90.5%) | 5.81(-82.2%) | 4.83(-85.2%) | 2.42(-92.6%) | 3.87(-88.1%) |
| 2 | r=1 | 10.21 | 3.66(-64.1%) | 2.49(-75.7%) | 0.71(-93.1%) | 0.81(-92.1%) | 1.18(-88.4%) | 0.89(-91.3%) | 0.66(-93.6%) | 1.07(-89.5%) |
| | r=5 | 17.69 | 7.64(-56.8%) | 8.78(-50.4%) | 1.91(-89.2%) | 2.26(-87.2%) | 3.61(-79.6%) | 2.47(-86.1%) | 1.79(-89.9%) | 2.87(-83.8%) |
| 4 | r=1 | 1.94 | 0.31(-84.2%) | 0.57(-70.8%) | 0.43(-78.0%) | 0.78(-59.9%) | 0.50(-74.0%) | 0.47(-75.8%) | 0.43(-77.9%) | 0.29(-85.0%) |
| | r=5 | 4.63 | 0.75(-83.9%) | 1.50(-67.6%) | 0.98(-78.8%) | 2.12(-54.3%) | 1.27(-72.5%) | 1.19(-74.3%) | 1.08(-76.8%) | 0.71(-84.6%) |
| 8 | r=1 | 0.73 | 0.20(-72.8%) | 0.34(-54.0%) | 0.32(-56.8%) | 0.61(-17.0%) | 0.38(-48.8%) | 0.32(-55.7%) | 0.32(-56.6%) | 0.22(-69.7%) |
| | r=5 | 1.93 | 0.44(-77.2%) | 0.83(-56.9%) | 0.68(-64.8%) | 1.59(-17.4%) | 0.87(-54.8%) | 0.74(-61.9%) | 0.73(-62.1%) | 0.51(-73.6%) |
| 16 | r=1 | 0.52 | 0.21(-60.0%) | 0.26(-49.9%) | 0.37(-28.3%) | 0.66(+26.7%) | 0.38(-28.1%) | 0.31(-40.3%) | 0.30(-42.5%) | 0.20(-62.4%) |
| | r=5 | 1.29 | 0.45(-64.8%) | 0.61(-52.7%) | 0.82(-36.4%) | 1.75(+35.7%) | 0.86(-32.9%) | 0.68(-46.9%) | 0.67(-47.6%) | 0.44(-66.1%) |
| 32 | r=1 | 0.34 | 0.17(-49.7%) | 0.20(-41.5%) | 0.35(+4.0%) | 0.62(+84.6%) | 0.36(+6.4%) | 0.31(-7.0%) | 0.28(-15.9%) | 0.17(-49.0%) |
| | r=5 | 0.80 | 0.35(-56.9%) | 0.42(-48.0%) | 0.76(-5.6%) | 1.64(+104.9%) | 0.82(+2.2%) | 0.68(-14.7%) | 0.62(-23.0%) | 0.37(-53.4%) |
| 64 | r=1 | 0.21 | 0.14(-34.4%) | 0.17(-22.5%) | 0.33(+54.2%) | 0.66(+210.8%) | 0.34(+60.6%) | 0.28(+31.0%) | 0.29(+35.4%) | 0.17(-19.2%) |
| | r=5 | 0.46 | 0.26(-43.4%) | 0.33(-29.3%) | 0.69(+49.8%) | 1.77(+282.5%) | 0.77(+66.0%) | 0.58(+26.3%) | 0.63(+36.7%) | 0.37(-19.0%) |
| 128 | r=1 | 0.23 | 0.14(-37.7%) | 0.17(-28.0%) | 0.32(+39.0%) | 0.65(+180.0%) | 0.33(+43.1%) | 0.29(+24.3%) | 0.28(+21.0%) | 0.17(-28.2%) |
| | r=5 | 0.50 | 0.28(-44.6%) | 0.33(-34.2%) | 0.66(+32.8%) | 1.69(+239.7%) | 0.73(+45.9%) | 0.59(+18.0%) | 0.60(+20.3%) | 0.36(-28.0%) |
| 256 | r=1 | 0.18 | 0.14(-24.5%) | 0.16(-13.9%) | 0.35(+88.4%) | 0.58(+211.9%) | 0.33(+80.9%) | 0.30(+61.6%) | 0.29(+59.1%) | 0.16(-12.9%) |
| | r=5 | 0.38 | 0.26(-32.2%) | 0.31(-20.1%) | 0.73(+91.5%) | 1.48(+286.7%) | 0.74(+91.6%) | 0.64(+67.4%) | 0.64(+67.2%) | 0.35(-9.3%) |
| 512 | r=1 | 0.19 | 0.12(-36.2%) | 0.17(-13.6%) | 0.36(+86.9%) | 0.59(+204.8%) | 0.34(+76.6%) | 0.29(+51.0%) | 0.30(+56.4%) | 0.17(-13.7%) |
| | r=5 | 0.41 | 0.23(-44.3%) | 0.33(-19.4%) | 0.78(+91.3%) | 1.54(+277.6%) | 0.76(+86.8%) | 0.61(+50.2%) | 0.68(+65.8%) | 0.36(-11.4%) |
| 1024 | r=1 | 0.20 | 0.14(-31.9%) | 0.19(-7.4%) | 0.33(+63.4%) | 0.66(+230.2%) | 0.34(+71.9%) | 0.29(+47.0%) | 0.30(+48.5%) | 0.16(-18.9%) |
| | r=5 | 0.42 | 0.25(-40.0%) | 0.37(-12.7%) | 0.68(+62.4%) | 1.75(+315.7%) | 0.76(+80.7%) | 0.62(+46.3%) | 0.65(+54.2%) | 0.35(-16.9%) |

Table 4: **Results for Diffusion** $\nu = 0.004$. Average $rL_2$ loss in percentage, and the relative change compared to C0 (no pretraining) in paranthesis. Negative changes indicate improvement.

| #samples | rollout | C0 | C1 | C2 | C3 | C4 | C5 | C6 | C7 | C8 |
|---|---|---|---|---|---|---|---|---|---|---|
| 1 | r=1 | 19.36 | 8.15(-57.9%) | 3.37(-82.6%) | 1.56(-92.0%) | 1.92(-90.1%) | 1.98(-89.8%) | 2.25(-88.4%) | 1.11(-94.3%) | 1.70(-91.2%) |
|  | r=5 | 34.60 | 16.93(-51.1%) | 11.68(-66.2%) | 4.19(-87.9%) | 5.04(-85.4%) | 6.29(-81.8%) | 6.53(-81.1%) | 2.89(-91.6%) | 4.54(-86.9%) |
| 2 | r=1 | 12.01 | 5.97(-50.3%) | 1.82(-84.9%) | 0.62(-94.9%) | 0.79(-93.4%) | 1.07(-91.1%) | 1.55(-87.1%) | 0.59(-95.1%) | 1.64(-86.3%) |
|  | r=5 | 19.50 | 11.20(-42.6%) | 5.42(-72.2%) | 1.47(-92.5%) | 1.97(-89.9%) | 3.05(-84.3%) | 4.76(-75.6%) | 1.39(-92.9%) | 4.10(-79.0%) |
| 4 | r=1 | 1.69 | 0.33(-80.7%) | 0.50(-70.2%) | 0.46(-72.9%) | 0.76(-55.0%) | 0.51(-69.8%) | 0.50(-70.6%) | 0.42(-75.0%) | 0.29(-83.1%) |
|  | r=5 | 3.92 | 0.71(-81.9%) | 1.21(-69.1%) | 0.93(-76.4%) | 1.81(-53.7%) | 1.13(-71.2%) | 1.11(-71.6%) | 0.90(-77.0%) | 0.60(-84.8%) |
| 8 | r=1 | 0.64 | 0.22(-65.0%) | 0.33(-48.2%) | 0.40(-38.1%) | 0.62(-2.6%) | 0.42(-33.7%) | 0.36(-44.5%) | 0.36(-43.3%) | 0.23(-63.6%) |
|  | r=5 | 1.62 | 0.42(-74.1%) | 0.72(-55.4%) | 0.76(-52.6%) | 1.42(-12.1%) | 0.88(-45.7%) | 0.70(-56.8%) | 0.71(-55.9%) | 0.45(-72.0%) |
| 16 | r=1 | 0.59 | 0.25(-57.5%) | 0.32(-46.1%) | 0.41(-30.2%) | 0.67(+13.9%) | 0.46(-22.2%) | 0.37(-37.1%) | 0.36(-38.8%) | 0.22(-62.3%) |
|  | r=5 | 1.37 | 0.49(-64.6%) | 0.68(-50.6%) | 0.80(-41.5%) | 1.58(+15.1%) | 0.97(-29.5%) | 0.73(-47.1%) | 0.71(-47.9%) | 0.42(-69.4%) |
| 32 | r=1 | 0.34 | 0.21(-39.0%) | 0.23(-33.8%) | 0.35(+3.0%) | 0.64(+85.7%) | 0.40(+17.5%) | 0.36(+6.2%) | 0.33(-4.5%) | 0.21(-38.6%) |
|  | r=5 | 0.72 | 0.36(-49.6%) | 0.42(-41.5%) | 0.63(-12.0%) | 1.47(+104.3%) | 0.80(+11.0%) | 0.69(-3.4%) | 0.62(-14.1%) | 0.38(-46.9%) |
| 64 | r=1 | 0.31 | 0.17(-44.6%) | 0.21(-31.8%) | 0.41(+33.4%) | 0.63(+105.0%) | 0.39(+25.1%) | 0.32(+2.8%) | 0.33(+7.2%) | 0.19(-37.6%) |
|  | r=5 | 0.61 | 0.26(-56.4%) | 0.37(-38.9%) | 0.78(+29.1%) | 1.45(+138.5%) | 0.74(+21.5%) | 0.57(-5.7%) | 0.62(+1.4%) | 0.34(-43.8%) |
| 128 | r=1 | 0.27 | 0.17(-36.2%) | 0.19(-28.5%) | 0.37(+37.4%) | 0.63(+131.3%) | 0.40(+45.6%) | 0.33(+22.3%) | 0.33(+21.2%) | 0.19(-28.8%) |
|  | r=5 | 0.52 | 0.27(-48.5%) | 0.33(-35.9%) | 0.67(+30.3%) | 1.44(+178.2%) | 0.77(+48.3%) | 0.61(+17.1%) | 0.61(+18.5%) | 0.35(-32.4%) |
| 256 | r=1 | 0.27 | 0.17(-36.7%) | 0.19(-28.1%) | 0.40(+47.9%) | 0.65(+141.9%) | 0.41(+52.4%) | 0.34(+26.4%) | 0.33(+22.8%) | 0.19(-30.9%) |
|  | r=5 | 0.50 | 0.26(-48.0%) | 0.32(-36.5%) | 0.73(+45.8%) | 1.50(+197.2%) | 0.81(+61.4%) | 0.63(+24.7%) | 0.62(+23.0%) | 0.33(-34.7%) |
| 512 | r=1 | 0.31 | 0.17(-44.8%) | 0.21(-34.2%) | 0.38(+20.4%) | 0.69(+121.1%) | 0.39(+25.0%) | 0.35(+11.5%) | 0.33(+3.9%) | 0.19(-39.6%) |
|  | r=5 | 0.61 | 0.27(-56.3%) | 0.35(-42.4%) | 0.69(+12.7%) | 1.63(+167.3%) | 0.76(+24.3%) | 0.64(+5.2%) | 0.60(-0.8%) | 0.34(-45.0%) |
| 1024 | r=1 | 0.27 | 0.19(-31.2%) | 0.22(-19.4%) | 0.38(+41.7%) | 0.69(+155.4%) | 0.41(+50.6%) | 0.35(+29.7%) | 0.32(+19.7%) | 0.20(-26.6%) |
|  | r=5 | 0.50 | 0.29(-41.6%) | 0.38(-24.4%) | 0.71(+40.0%) | 1.63(+222.1%) | 0.80(+58.5%) | 0.64(+26.3%) | 0.60(+18.7%) | 0.35(-30.1%) |

Table 5: **Results for Advection** $\beta = 0.1$. Average $rL_2$ loss in percentage, and the relative change compared to C0 (no pretraining) in paranthesis. Negative changes indicate improvement.

| #samples | rollout | C0 | C1 | C2 | C3 | C4 | C5 | C6 | C7 | C8 |
|---|---|---|---|---|---|---|---|---|---|---|
| 1 | r=1 | 57.53 | 55.49(-3.5%) | 32.18(-44.1%) | 12.92(-77.5%) | 7.91(-86.3%) | 14.71(-74.4%) | >100 | 18.86(-67.2%) | 25.87(-55.0%) |
|  | r=5 | >100 | >100 | >100 | 24.72(-100.0%) | 17.97(-100.0%) | 37.25(-100.0%) | >100 | 33.92(-100.0%) | >100 |
| 2 | r=1 | 34.24 | 54.90(+60.3%) | 22.57(-34.1%) | 7.38(-78.4%) | 4.72(-86.2%) | 15.02(-56.1%) | >100 | 13.83(-59.6%) | 14.31(-58.2%) |
|  | r=5 | >100 | >100 | >100 | 14.21(-100.0%) | 10.16(-100.0%) | 47.16(-100.0%) | >100 | 23.62(-100.0%) | >100 |
| 4 | r=1 | 3.67 | 2.39(-34.9%) | 2.57(-30.1%) | 2.50(-31.8%) | 2.88(-21.7%) | 2.55(-30.7%) | 2.74(-25.4%) | 2.43(-33.9%) | 2.30(-37.4%) |
|  | r=5 | 7.84 | 4.37(-44.3%) | 5.09(-35.1%) | 4.36(-44.4%) | 5.92(-24.5%) | 4.49(-42.7%) | 5.45(-30.5%) | 4.15(-47.1%) | 4.06(-48.2%) |
| 8 | r=1 | 2.77 | 2.51(-9.7%) | 2.32(-16.2%) | 2.37(-14.5%) | 2.71(-2.4%) | 2.41(-13.1%) | 2.55(-8.2%) | 2.33(-16.1%) | 2.16(-22.0%) |
|  | r=5 | 5.56 | 4.31(-22.6%) | 4.46(-19.8%) | 3.97(-28.6%) | 5.45(-2.1%) | 4.19(-24.7%) | 4.80(-13.7%) | 3.97(-28.6%) | 3.79(-31.9%) |
| 16 | r=1 | 1.95 | 1.90(-2.7%) | 1.93(-1.3%) | 2.32(+19.0%) | 2.73(+39.8%) | 2.37(+21.2%) | 2.13(+9.1%) | 2.18(+11.6%) | 1.90(-2.7%) |
|  | r=5 | 3.31 | 2.92(-11.8%) | 3.19(-3.5%) | 3.67(+10.8%) | 5.42(+63.7%) | 3.86(+16.7%) | 3.76(+13.4%) | 3.51(+6.0%) | 3.00(-9.4%) |
| 32 | r=1 | 1.75 | 1.71(-2.3%) | 1.76(+0.3%) | 2.26(+28.7%) | 2.59(+47.8%) | 2.32(+32.5%) | 1.96(+12.2%) | 2.22(+26.7%) | 1.84(+5.1%) |
|  | r=5 | 2.61 | 2.39(-8.3%) | 2.60(-0.2%) | 3.47(+33.0%) | 5.00(+91.6%) | 3.66(+40.2%) | 3.24(+24.2%) | 3.44(+31.9%) | 2.79(+6.8%) |
| 64 | r=1 | 1.67 | 1.70(+1.9%) | 1.73(+3.9%) | 2.28(+37.0%) | 2.67(+60.4%) | 2.33(+40.1%) | 2.20(+32.0%) | 2.25(+34.9%) | 1.79(+7.8%) |
|  | r=5 | 2.27 | 2.29(+0.6%) | 2.45(+7.6%) | 3.46(+52.0%) | 5.19(+128.3%) | 3.64(+60.0%) | 3.47(+52.5%) | 3.44(+51.1%) | 2.65(+16.6%) |
| 128 | r=1 | 1.65 | 1.64(-0.3%) | 1.69(+2.7%) | 2.25(+36.5%) | 2.68(+62.5%) | 2.36(+43.2%) | 2.15(+30.7%) | 2.25(+36.6%) | 1.84(+11.8%) |
|  | r=5 | 2.19 | 2.13(-2.9%) | 2.32(+5.8%) | 3.39(+54.9%) | 5.22(+137.9%) | 3.71(+69.2%) | 3.38(+54.1%) | 3.41(+55.6%) | 2.73(+24.6%) |
| 256 | r=1 | 1.65 | 1.66(+0.3%) | 1.67(+1.1%) | 2.29(+38.7%) | 2.70(+63.2%) | 2.35(+42.2%) | 2.18(+32.1%) | 2.22(+34.1%) | 1.80(+8.9%) |
|  | r=5 | 2.17 | 2.17(-0.3%) | 2.25(+3.5%) | 3.47(+59.5%) | 5.26(+141.7%) | 3.69(+69.6%) | 3.41(+56.7%) | 3.34(+53.7%) | 2.65(+22.0%) |
| 512 | r=1 | 1.60 | 1.65(+3.0%) | 1.69(+5.4%) | 2.27(+42.0%) | 2.67(+66.9%) | 2.31(+44.6%) | 2.11(+32.1%) | 2.23(+39.3%) | 1.79(+11.9%) |
|  | r=5 | 2.03 | 2.15(+5.5%) | 2.30(+12.9%) | 3.40(+67.1%) | 5.20(+155.6%) | 3.59(+76.5%) | 3.28(+61.0%) | 3.38(+66.0%) | 2.63(+29.1%) |
| 1024 | r=1 | 1.62 | 1.65(+1.7%) | 1.68(+3.5%) | 2.29(+41.4%) | 2.65(+63.8%) | 2.31(+42.9%) | 2.21(+36.4%) | 2.27(+40.1%) | 1.78(+9.7%) |
|  | r=5 | 2.09 | 2.13(+2.0%) | 2.26(+8.2%) | 3.45(+64.9%) | 5.14(+145.9%) | 3.57(+70.7%) | 3.42(+63.6%) | 3.42(+63.5%) | 2.59(+23.7%) |

Table 6: **Results for Advection** $\beta = 0.4$. Average $rL_2$ loss in percentage, and the relative change compared to C0 (no pretraining) in paranthesis. Negative changes indicate improvement.

| #samples | rollout | C0 | C1 | C2 | C3 | C4 | C5 | C6 | C7 | C8 |
|---|---|---|---|---|---|---|---|---|---|---|
| 1 | r=1 | 33.36 | 34.54(+3.5%) | 29.75(-10.8%) | 13.20(-60.4%) | 26.86(-19.5%) | 18.37(-44.9%) | 78.35(+134.9%) | 18.59(-44.3%) | 26.81(-19.6%) |
|  | r=5 | >100 | >100 | >100 | >100 | >100 | 41.04(-100.0%) | >100 | 46.89(-100.0%) | >100 |
| 2 | r=1 | 29.71 | 27.36(-7.9%) | 27.05(-9.0%) | 9.68(-67.4%) | 20.77(-30.1%) | 13.25(-55.4%) | >100 | 12.99(-56.3%) | 20.74(-30.2%) |
|  | r=5 | >100 | >100 | >100 | >100 | >100 | 23.40(-100.0%) | >100 | 48.15(-100.0%) | >100 |
| 4 | r=1 | 5.27 | 4.11(-21.9%) | 4.37(-17.0%) | 4.83(-8.2%) | 14.09(+167.6%) | 4.93(-6.3%) | 4.52(-14.1%) | 4.50(-14.6%) | 8.91(+69.3%) |
|  | r=5 | 11.38 | 11.34(-0.3%) | 8.99(-21.0%) | 11.71(+2.9%) | >100 | 8.96(-21.3%) | 9.92(-12.8%) | 8.81(-22.6%) | 24.57(+115.9%) |
| 8 | r=1 | 3.91 | 3.46(-11.6%) | 3.65(-6.8%) | 3.90(-0.2%) | 12.71(+225.1%) | 4.45(+13.9%) | 3.83(-2.1%) | 3.81(-2.5%) | 6.40(+63.7%) |
|  | r=5 | 9.07 | 6.48(-28.6%) | 10.88(+19.9%) | 8.68(-4.4%) | 41.73(+360.0%) | 7.85(-13.5%) | 8.52(-6.1%) | 7.44(-18.0%) | 16.78(+84.9%) |
| 16 | r=1 | 2.59 | 2.60(+0.3%) | 2.72(+5.0%) | 3.37(+30.3%) | 11.64(+349.8%) | 4.32(+66.8%) | 3.17(+22.4%) | 3.46(+33.6%) | 4.73(+82.9%) |
|  | r=5 | 4.55 | 4.29(-5.6%) | 4.95(+8.7%) | 6.58(+44.5%) | 38.78(+752.0%) | 7.16(+57.3%) | 6.31(+38.6%) | 6.44(+41.4%) | 11.91(+161.6%) |
| 32 | r=1 | 2.11 | 2.09(-0.9%) | 2.24(+6.3%) | 3.15(+49.3%) | 11.14(+427.9%) | 4.25(+101.6%) | 2.91(+38.0%) | 3.29(+56.1%) | 3.96(+87.9%) |
|  | r=5 | 3.32 | 3.10(-6.7%) | 3.60(+8.6%) | 5.51(+66.1%) | 37.01(+1015.3%) | 6.60(+98.9%) | 5.54(+66.9%) | 5.57(+68.0%) | 9.56(+188.1%) |
| 64 | r=1 | 1.94 | 1.90(-2.0%) | 2.05(+5.5%) | 3.00(+54.9%) | 11.46(+491.0%) | 4.25(+119.1%) | 2.76(+42.3%) | 3.58(+84.5%) | 3.98(+105.0%) |
|  | r=5 | 2.73 | 2.58(-5.3%) | 2.98(+9.2%) | 5.12(+87.8%) | 36.71(+1245.0%) | 6.60(+142.0%) | 5.10(+86.8%) | 5.79(+112.1%) | 9.89(+262.3%) |
| 128 | r=1 | 1.84 | 1.92(+4.5%) | 1.95(+5.7%) | 3.13(+69.9%) | 11.41(+520.0%) | 4.23(+130.1%) | 2.80(+51.9%) | 3.24(+76.0%) | 3.62(+96.6%) |
|  | r=5 | 2.43 | 2.54(+4.5%) | 2.66(+9.2%) | 5.30(+117.6%) | 36.00(+1379.2%) | 6.39(+162.4%) | 5.13(+110.7%) | 5.28(+116.8%) | 8.54(+250.9%) |
| 256 | r=1 | 1.88 | 1.93(+2.6%) | 1.91(+1.8%) | 3.22(+71.3%) | 11.49(+511.1%) | 4.31(+129.2%) | 2.94(+56.3%) | 3.72(+97.5%) | 3.84(+104.4%) |
|  | r=5 | 2.51 | 2.53(+0.5%) | 2.55(+1.5%) | 5.67(+125.6%) | 36.73(+1361.1%) | 6.49(+158.2%) | 5.42(+115.5%) | 5.94(+136.3%) | 9.50(+277.8%) |
| 512 | r=1 | 1.75 | 1.82(+4.3%) | 1.92(+10.1%) | 3.23(+84.5%) | 11.57(+561.5%) | 4.29(+145.3%) | 2.73(+56.3%) | 3.15(+80.2%) | 3.84(+119.6%) |
|  | r=5 | 2.19 | 2.31(+5.5%) | 2.53(+15.5%) | 5.42(+147.0%) | 35.94(+1538.9%) | 6.62(+202.0%) | 4.98(+127.2%) | 5.27(+140.4%) | 9.57(+336.4%) |
| 1024 | r=1 | 1.80 | 1.84(+2.2%) | 1.87(+4.2%) | 3.04(+69.2%) | 11.42(+535.5%) | 4.31(+140.0%) | 2.72(+51.2%) | 3.47(+93.2%) | 3.67(+104.2%) |
|  | r=5 | 2.31 | 2.32(+0.4%) | 2.42(+4.8%) | 4.86(+110.2%) | 35.97(+1457.0%) | 6.60(+185.6%) | 4.89(+111.5%) | 5.64(+144.2%) | 8.85(+283.3%) |

Table 7: **Results for Advection** $\beta = 1.0$. Average $rL_2$ loss in percentage, and the relative change compared to C0 (no pretraining) in paranthesis. Negative changes indicate improvement.

| #samples | rollout | C0 | C1 | C2 | C3 | C4 | C5 | C6 | C7 | C8 |
|---|---|---|---|---|---|---|---|---|---|---|
| 1 | r=1 | 33.41 | 34.49(+3.2%) | 24.32(-27.2%) | 25.86(-22.6%) | 91.66(+174.4%) | 26.89(-19.5%) | 27.02(-19.1%) | 26.96(-19.3%) | 65.14(+95.0%) |
|  | r=5 | >100 | >100 | >100 | >100 | >100 | 85.98(-66.5%) | >100 | >100 | >100 |
| 2 | r=1 | 24.63 | 34.99(+42.1%) | 14.79(-40.0%) | 16.15(-34.4%) | 78.68(+219.4%) | 16.83(-31.7%) | 38.39(+55.9%) | 16.86(-31.6%) | 41.10(+66.8%) |
|  | r=5 | >100 | >100 | >100 | 88.24(-100.0%) | >100 | 66.09(-100.0%) | >100 | 31.88(-100.0%) | >100 |
| 4 | r=1 | 4.76 | 4.04(-15.1%) | 4.38(-8.1%) | 6.86(+44.0%) | 57.32(+1103.7%) | 5.64(+18.4%) | 4.93(+3.5%) | 5.57(+16.9%) | 22.51(+372.8%) |
|  | r=5 | 10.78 | 9.01(-16.4%) | 10.39(-3.6%) | 83.19(+671.8%) | >100 | 14.37(+33.3%) | 13.12(+21.7%) | 12.74(+18.2%) | 71.41(+562.5%) |
| 8 | r=1 | 3.49 | 3.23(-7.4%) | 3.47(-0.6%) | 4.99(+42.8%) | 47.71(+1265.9%) | 4.68(+34.1%) | 4.03(+15.4%) | 4.55(+30.1%) | 14.17(+305.8%) |
|  | r=5 | 7.80 | 6.92(-11.3%) | 7.74(-0.8%) | 19.24(+146.6%) | >100 | 11.24(+44.1%) | 10.16(+30.3%) | 9.91(+27.0%) | 46.89(+501.0%) |
| 16 | r=1 | 2.48 | 2.38(-3.9%) | 2.66(+7.4%) | 3.85(+55.6%) | 40.02(+1515.2%) | 4.56(+83.9%) | 3.50(+41.1%) | 4.13(+66.5%) | 10.44(+321.5%) |
|  | r=5 | 4.79 | 4.53(-5.5%) | 5.46(+14.0%) | 13.85(+189.1%) | >100 | 10.57(+120.6%) | 8.32(+73.5%) | 8.69(+81.4%) | 31.74(+562.5%) |
| 32 | r=1 | 2.10 | 2.10(-0.1%) | 2.21(+5.3%) | 3.25(+54.6%) | 36.02(+1613.0%) | 4.39(+108.7%) | 3.18(+51.1%) | 3.69(+75.3%) | 8.82(+319.5%) |
|  | r=5 | 3.62 | 3.56(-1.7%) | 4.05(+11.9%) | 9.23(+155.0%) | >100 | 10.04(+177.4%) | 7.23(+99.7%) | 7.35(+103.2%) | 30.69(+747.9%) |
| 64 | r=1 | 1.99 | 1.92(-3.2%) | 2.07(+4.2%) | 3.32(+67.0%) | 35.28(+1675.3%) | 4.37(+120.1%) | 3.30(+65.9%) | 3.77(+89.6%) | 7.59(+282.1%) |
|  | r=5 | 3.10 | 2.94(-5.1%) | 3.44(+11.1%) | 9.37(+202.4%) | >100 | 10.01(+223.1%) | 7.50(+142.1%) | 7.50(+142.1%) | 25.69(+729.2%) |
| 128 | r=1 | 1.88 | 1.98(+5.2%) | 2.02(+7.5%) | 3.07(+62.8%) | 32.71(+1636.9%) | 4.34(+130.3%) | 3.19(+69.4%) | 3.76(+99.4%) | 7.85(+317.0%) |
|  | r=5 | 2.81 | 2.99(+6.4%) | 3.23(+15.1%) | 7.82(+178.7%) | >100 | 9.96(+255.1%) | 7.27(+158.9%) | 7.44(+165.1%) | 27.61(+883.9%) |
| 256 | r=1 | 1.86 | 1.93(+4.0%) | 1.92(+3.4%) | 3.03(+63.4%) | 35.44(+1810.0%) | 4.11(+121.6%) | 3.10(+67.3%) | 3.81(+105.2%) | 8.15(+339.3%) |
|  | r=5 | 2.69 | 2.85(+5.9%) | 2.88(+7.2%) | 6.65(+147.3%) | >100 | 9.16(+240.5%) | 6.96(+158.8%) | 7.58(+181.7%) | 32.01(+1090.4%) |
| 512 | r=1 | 1.85 | 1.93(+4.3%) | 1.93(+4.3%) | 3.03(+63.8%) | 35.44(+1814.2%) | 4.14(+123.6%) | 3.09(+66.8%) | 3.62(+95.6%) | 7.72(+317.1%) |
|  | r=5 | 2.68 | 2.82(+5.1%) | 2.94(+9.5%) | 6.68(+149.0%) | >100 | 9.31(+247.0%) | 6.86(+155.7%) | 7.01(+161.4%) | 27.68(+932.1%) |
| 1024 | r=1 | 1.85 | 1.90(+2.2%) | 1.98(+6.6%) | 3.10(+67.2%) | 36.42(+1863.8%) | 4.30(+131.7%) | 3.40(+83.1%) | 3.63(+95.8%) | 7.71(+315.6%) |
|  | r=5 | 2.67 | 2.75(+3.0%) | 3.02(+13.2%) | 7.57(+184.0%) | >100 | 9.75(+265.6%) | 7.88(+195.6%) | 7.11(+166.5%) | 28.72(+976.8%) |

