# OpenReview forum: "Pretraining a Neural Operator in Lower Dimensions"
_TMLR — Accepted by TMLR_

### Review · Reviewer_YiZw · 2024-09-26

**Summary Of Contributions:**

This paper evaluates the effectiveness of pretraining for learning neural PDE solvers. Due to the cost of acquiring quality simulations of high dimensional PDEs, the authors suggest pretraining neural PDE solvers with 1D PDEs and fine-tuning them on 2D downstream tasks. Amongst the selected list of fine-tuning configurations, fine-tuning all parameters (C1) appear to work best.

**Audience:**

Yes

**Broader Impact Concerns:**

I have no concerns on the ethical implications at this stage.

**Claims And Evidence:**

No

**Requested Changes:**

1. The authors mention the need for exponential samples for fine-discretizations of higher dimensional PDEs but they are no hard numbers. For eg, the downstream relative error of the best-fine-tuned model is about 20%. How should one perceive this value? Does the potentially costly 2D data acquisition drive the relative error of the full-trained models to 0%? If so for how much sample data? What is the trade-off between pretaining-vs-finetuning?

If one cares about achieving relative error at most $\epsilon$, is this achievable with only pertaining? or is data acquisition necessary? Even if proving sample complexity theorems on the data-ratios needed for pertaining is non-trivial, There should be a more comprehensive evaluation of observed empirical sample complexity since the data distributions here are synthetic mixtures derived from well founded PDE equations.


2. There are some text style issues with citation (check for repeated or mishaped citations).

**Strengths And Weaknesses:**

__Strengths__
There is much value in establishing the best practices on pretraining-vs-finetuning for neural PDE solvers. This work undertakes preliminary peek at this core issue.

__Weakness__
The best recipe for balancing pretraining-vs-finetuning depends on the cost of data acquisition, training compute resources, model scaling, and sample complexity of downstream tasks. There is insufficient rigorous discussion on each of these facets.

---

> ### Author Response · Authors · 2024-10-18
> **Response to reviewer YiZw**
>
> There seems to be some misunderstanding about some claims and statements in the article, and the procedure and justifications of pretraining. Therefore, we would like to provide some clarification about the queried topics by this reviewer. We greatly appreciate the reviewer’s time and effort and look forward to further feedback to improve the clarity of the claims of the article and include necessary information.
>
> First, _we have not made any claim about the number of needed samples for training the models, or it being exponential. The statements are about the cost of numerical methods or neural network training and inference with the increase of the number of dimensions._ The statements about the cost of data collection and model training on high-dimensional PDEs is to mainly highlight the lower cost of our pretraining strategy compared to other pretraining strategies, not the fine-tuning.
>
> Second, pretraining and fine-tuning are not in a rivalry, where a certain amount of one would be equivalent to some amount of the other in terms of achieving a certain performance. Pretraining is the training of the model on another task, which may use another dataset or the same dataset with a different learning task, before training the model on the main task and dataset. Fine-tuning is the training of the same model, starting from where it left off at the end of the pretraining stage, to learn the main/downstream task. Therefore, there is no notion of a balance or trade-off between pretraining and fine-tuning, i.e. pretraining-vs-finetuning, as _the two stages are both executed until convergence to get the best result possible. The low cost and availability of the pretraining data can present a strong motivation, but it is not the only one._ Sometimes the same exact data with a differently designed learning task is used in pretraining, because _pretraining has the potential to prepare the parameters for the main task to achieve a better final performance, which may not be possible by simply acquiring more data for the main task._  That is why pretraining is still popular and applied even where the cost of data collection and pretraining is high. Therefore, _the metric to look out for is the performance gain of pretrained model compared to the one without pretraining,_ which are explicitly included in parenthesis in the appendix tables. The downstream error itself is not indicative of the success of pretraining, as training directly on the downstream task may also be able to achieve that error. We have explicitly provided the gain of pretrained models over non-pretrained ones in the appendix, and a few times in the text.
>
> Third, the notion of sample complexity, i.e. the guaranteed probabilistic relationship between the amount of training data and the achieved test error, is usually discussed in machine learning theory for shallow ML models such as linear or logistic regression. It usually is not and arguably cannot be applied to deep learning models. Even empirically speaking, this perspective is not usually taken in related works in this field due to the complicated distribution of the data and model architectures. To provide a particular answer to the questions, _there is no amount of data, either with or without pretraining, that can guarantee a certain test error like ϵ or 0_, as the result depends on the random initialization of the parameters in a very high-dimensional space of the network parameters and making claims about the average outcome would be too uncertain. The error of zero is practically never achieved by such models, because the test dataset is assumed to contain unseen data. Still, _we have executed our experiments for a wide range of available downstream data to investigate the trend and the relationship of available downstream training data and the final test error._ It is observed in the plots that the test error of the model stops showing significant improvement after the available training data reaches a certain amount (more downstream training data does not lead to more error reduction after that point). The pretrained model has a similar behavior, but with a better asymptotic final error, thanks to the pretraining. This is what makes the case for the effectiveness of this pretraining strategy.
>
> __APPLIED CHANGES:__
>
> In the new version of the article, the second section is related works. In the last two paragraphs, we have tried our best to make the case for our pretraining strategy and make the writing for clear and reduce confusion. At the end of the second last paragraph, we have provided a more formal quantification of the cost comparison of 1D and 2D PDEs.
>
> We have also included the training time on 1D and 2D data for our specific experiments at the end of subsection 5.1.
>
> We have also fixed the citation style issue. We apologize for the styling mistake.

---

### Review · Reviewer_X8Wp · 2024-09-27

**Summary Of Contributions:**

Summary:  The authors propose a pretraining approach for neural operator-based models to increase the accuracy and training efficiency of a neural operator for high-dimensional PDEs. The strategy is simple, i.e., to use a pre-trained model in lower dimensions and fine-tune it in higher dimensions.

**Audience:**

Yes

**Claims And Evidence:**

Yes

**Requested Changes:**

Weaknesses / Requested Changes
I do not have any major criticisms of the paper since the goal seems pretty straightforward. However, I have some minor comments:

1.The pretraining strategy proposed seems to be compatible only with FFNO-like operators and, hence, does not seem like a generic approach for solver fine-tuning. It would be great if the authors could elaborate more on this limitation of their approach.

2. How were the combinations in Table 1 chosen, and what is the motivation behind such a selection? I would request the authors to clarify the experimental setup with motivation here.

3. Throughout the paper, citations are in line with the text without a delimiter like () which makes reading at times very tedious. I would request the authors to fix this problem.

**Strengths And Weaknesses:**

Strengths:

1. The method is straightforward, and the paper is well-written in terms of discussing existing work on operators and finetuning operators for downstream tasks.
2. The empirical results show that the proposed method can be useful for some PDEs, like the diffusion equation. As would be expected the method works well than simple pre-training in the low data regime and converges to that of the performance of directly pre-training on the downstream tasks for more sample budget.

---

> ### Author Response · Authors · 2024-10-18
> **Response to reviewer X8Wp**
>
> We would like to thank the reviewer for their concise summary and feedback. We have applied the following changes in the revised version according to the provided comments. We would appreciate your follow-up feedback on any remaining modifications we can apply to improve the article further.
>
> 1- We have added a new paragraph and a pseudocode at the beginning of the methodology section, which elaborates on this requirement more explicitly.
>
> 2- It is already included in the article that we are taking a different fine-tuning approach from that of classical deep learning applications, and that it is based on the type and location of the modules in the model. To further elaborate on the specific choices in our experiments, we have added the following text in the methodology section:
>
> _The configurations C2 to C8 were selected so the tunable modules are either in proximity to the model's input or output, or are of the same type (Fourier or feedforward). While C1 represents a fully tunable model, C2 and C3 have a certain tunable layer type across all layers and the remaining configurations have one or both layer types from the first and/or last layer._
>
> 3- We have fixed the citation style issue. We apologize for the mistake.

---

### Review · Reviewer_ksvx · 2024-10-05

**Summary Of Contributions:**

Pretraining via proxy tasks has been explored for neural PDE solvers, but most of the existing works relied on a large pretraining dataset with a high collection cost. In this paper, the authors presents the approach of pretraining neural PDE solvers on lower dimensional PDEs (PreLowD), and argues that 1D PDE data is significantly lower than high-dimensional PDE data. In the experiments, the authors pretrain fine-tuning factorized Fourier neural operators on 10K 1D PDE samples, and then fine-tune them on two 2D PDEs (diffusion and advection) to show that they outperform neural operators trained from scratch.

**Audience:**

Yes

**Broader Impact Concerns:**

There are no significant ethical concerns related to the broader impact of this work.

**Claims And Evidence:**

No

**Requested Changes:**

See the weaknesses above. It would be good if the authors could (1) improve the writing quality and (2) include more experiments to justify their approach.

**Strengths And Weaknesses:**

Strengths
* The approach of pretraining then fine-tuning has led to the huge success of LLMs. It is interesting to explore how to adapt this strategy to solving PDEs. This paper particularly studies one of the key bottlenecks: the lack of data.
* The PreLowD approach demonstrates significant error reduction compared to training from scratch.

Weaknesses:

* This paper's writing quality could be significantly improved. Some sections are hard to follow.
  1. Introduction: Much of the content of the introduction could be moved to related works. The main contribution of the paper is mentioned in the last paragraph, but it is better to also describe the approach with slightly more details and highlight some key experiment results.
  2. Methodology: This is the most confusing part. The proposed approach should have been described in detail here, but it is very hard to see what the PreLowD approach really is through its current presentation. It is better to describe the approach step by step, and mention what type of models, data and training procedures should be used in each step. In fact, even after reading the entire paper, I still do not know what the pretraining PDEs are.
* Experiments are not thorough enough.
  1. The pretrained neural operators were only tested on two 2D PDEs. It would be better to test them on more PDEs.
  2. This paper proposes 8 fine-tuning approaches in a row. It would be better to design some metrics (such as the average loss over downstream tasks) to compare them thoroughly and make a recommendation for one particular approach.
  3. The pretraining dataset seems to be constructed a bit arbitrarily. Can the authors provide some ablation studies on what data should be included in pretraining?
* The paper mentioned that 1D PDE data is easier to generate. Why is this the case? It would be better to write a paragraph to compare the costs of generating 1D and 2D PDE data.

Typos:
* Section 1.2: Subramanian et al. Subramanian et al. (2024)
* Section 2.2: Li et al. Li et al. (2020b)

---

> ### Author Response · Authors · 2024-10-18
> **Response to reviewer ksvx**
>
> We would like to greatly thank this reviewer as their feedback has helped us improve the writing quality of the paper and detect and fix several potential causes of misunderstanding and confusion in the paper. Please find the corresponding responses to each comment below. We look forward to the constructive feedback to further improve the article.
>
> - Writing quality:
>
>     1. Thank you for this constructive feedback. We have changed the architecture of the paper such that the literature review on pretraining neural solvers is now under Related works, and we have included the following about the results of our experiments at the end of the related works section.
>     2. We have added a new paragraph and a pseudocode at the beginning of the methodology section for a formal and explicit definition of the approach.
>
> - Experiments:
>     1. We agree that executing more experiments with more datasets and tasks is always better, but the experiments for this framework were very extensive and time-consuming since the pretrained models are to be fine-tuned with different amounts of downstream training data, different fine-tuning configurations, and different random seeds. Moreover, it is not reasonable to expect a successful fine-tuning on any arbitrary test PDE. The target PDE needs to have reasonable relevance to the pretraining PDE so that transfer learning is justified. Our choices of pretraining and downstream PDEs are guaranteed to have this relevance (since they are the same equations and phenomena, just with different dimensionalities) and represent two fundamental modes of transport phenomena. We acknowledge that our results do not guarantee the success of the framework for all possible applications, as is the case for virtually all deep learning experiments. Rather, it sheds light on a particular pretraining strategy that is possible for certain neural solvers thanks to their special architecture. Future research can investigate more complicated PDEs and advanced prelowding strategies such as training different layers in different stages on single terms or phenomena in lower dimensions. To conclude, further experiments exceed the time limit for this revision, and we believe it is best to leave them for future works.
>
>     2. This work investigates if prelowded models can generalize what they learn about a certain PDE in a low-dimensional domain to a higher-dimensional domain, with the same or similar governing equation or physical phenomena. The practice of evaluating fine-tuned models on a variety of downstream tasks is common in computer vision and language processing due to the hierarchical model architecture and learned features in those fields and the ability of the low-level features to be used in different tasks. We have argued in the paper that neural PDE solvers do not necessarily have the same properties, so we do not study them in the same way here.
>
>     For the case of the relevant downstream task, the final error on the test dataset of the downstream task is reported for each configuration, each PDE (advection or diffusion), and each value of the PDE coefficient, over two different prediction horizons (1 and 5 steps). The exact relative improvement for each case can also be found in the appendix tables. Since the gain depends on many factors, we believe it is best to avoid a collective metric of all experiments and recommending a certain configuration. The decision should be made conditionally on the available data and the task at hand. However, in the successful case (diffusion), the winner is almost consistent across all experiments (C1 and C8), so a collective metric would not present new information to supposedly choose the one with the most number of situations being the winner.
>
>     To summarize, we observe that the best approach depends on the amount of available data. However, unless the downstream data is extremely scarce, it is recommended that C1, i.e. letting all the parameters be fine-tuned, is the best fine-tuning approach in general. We have included an additional sentence at the end of the second paragraph of the conclusion for further clarification about the general recommendation.
>
> 3. We apologize for the confusion and unclarity about the dataset setup, which we believe was caused by this sentence:
> _We use the entire 1D training dataset in the pretraining stage but use a limited subset of the 2D training data for the downstream task._
> The pretraining datasets are not and should not be arbitrarily constructed. For each downstream 2D PDE, the pretraining dataset consists of samples from the relevant 1D PDE with the same coefficient. We also use the 1D and 2D version of the same formula for the generation of the initial conditions of the 1D and 2D datasets. The second paragraph of subsection 5.1 is now rewritten to prevent this confusion. We would like to greatly thank the reviewer for catching this error with their comment.

---

### Decision · Action_Editor_ELuq · 2024-11-13

**Recommendation:** Accept as is

**Comment:**

The reviewers unanimously voted "Leaning Accept". The core issues presented by reviewers were:
  * Clarity of writing and readability - this has been resolved.
  * Limited novelty - currently the method is only applicable to FFNO / a specific set of neural operators.
  * Limited experimental ablations - this has not been fully resolved, i.e. the paper performs experiments only on two specific types of PDEs and there are many questions raised around model scaling / sample complexity, etc.

**Audience:**

Yes - the paper presents preliminary research on pretraining neural PDE solvers over low dimensional data and then finetuning over high dimensional data, to reduce costs. Would be of interest to the neural ODE/PDE community.

**Claims And Evidence:**

Yes, despite the reviewer comments on limited novelty and lack of extensive studies, the paper therefore is quite careful around its claims and contributions. Page 3, first paragraph specifically states exactly:
  * What is being experimented with (Two well-known PDEs, diffusion and advection, in 1D and 2D)
  * Results (pretraining in 1D leads to lower errors in 2D than simply training purely in 2D)
  * Limitations (not necessarily applicable to any arbitrary PDE, but currently those which have a Factorized Fourier representation)